# Long-Term Task- and Dopamine-Dependent Dynamics of Subthalamic Local Field Potentials in Parkinson’s Disease

**DOI:** 10.3390/brainsci6040057

**Published:** 2016-11-29

**Authors:** Sara J. Hanrahan, Joshua J. Nedrud, Bradley S. Davidson, Sierra Farris, Monique Giroux, Aaron Haug, Mohammad H. Mahoor, Anne K. Silverman, Jun Jason Zhang, Adam Olding Hebb

**Affiliations:** 1Colorado Neurological Institute, Englewood, CO 80113, USA; shanrahan@thecni.org (S.J.H.); jnedrud@thecni.org (J.J.N.); 2Department of Mechanical and Materials Engineering, University of Denver, Denver, CO 80208, USA; Bradley.Davidson@du.edu; 3Movement and Neuroperformance Center of Colorado, Englewood, CO 80113, USA; sierrafarris@gmail.com (S.F.); giroux_ml@yahoo.com (M.G.); 4Blue Sky Neurology, Englewood, CO 80113, USA; aaronhaug@gmail.com; 5Department of Electrical and Computer Engineering, University of Denver, CO 80208, USA; Mohammad.Mahoor@du.edu (M.H.M.); Jun.Zhang@du.edu (J.J.Z.); 6Department of Mechanical Engineering, Colorado School of Mines, Golden, CO 80401, USA; asilverm@mines.edu

**Keywords:** Parkinson’s disease (PD), local field potential (LFP), deep brain stimulation (DBS), beta frequency oscillations, subthalamic nucleus, closed-loop

## Abstract

Subthalamic nucleus (STN) local field potentials (LFP) are neural signals that have been shown to reveal motor and language behavior, as well as pathological parkinsonian states. We use a research-grade implantable neurostimulator (INS) with data collection capabilities to record STN-LFP outside the operating room to determine the reliability of the signals over time and assess their dynamics with respect to behavior and dopaminergic medication. Seven subjects were implanted with the recording augmented deep brain stimulation (DBS) system, and bilateral STN-LFP recordings were collected in the clinic over twelve months. Subjects were cued to perform voluntary motor and language behaviors in on and off medication states. The STN-LFP recorded with the INS demonstrated behavior-modulated desynchronization of beta frequency (13–30 Hz) and synchronization of low gamma frequency (35–70 Hz) oscillations. Dopaminergic medication did not diminish the relative beta frequency oscillatory desynchronization with movement. However, movement-related gamma frequency oscillatory synchronization was only observed in the medication on state. We observed significant inter-subject variability, but observed consistent STN-LFP activity across recording systems and over a one-year period for each subject. These findings demonstrate that an INS system can provide robust STN-LFP recordings in ambulatory patients, allowing for these signals to be recorded in settings that better represent natural environments in which patients are in a variety of medication states.

## 1. Introduction

The subthalamic nucleus (STN) is a common target for deep brain stimulation (DBS) therapy in patients with Parkinson’s disease (PD). DBS alleviates the motor symptoms of PD [1], but may lead to side effects, such as impaired cognition [2], speech [3,4], gait [5] and balance [6]. An adaptive DBS system with active modulation of stimulation by appropriate physiological control variables may reduce these side effects, while providing maximum therapeutic benefit of PD motor symptoms [7,8,9,10,11]. Local field potentials (LFP) recorded from the STN have the potential to be a robust control signal to indicate a change in a patient’s state [12,13,14,15]. STN-LFPs have been shown to correlate with a patient’s Parkinson’s disease symptom state [16], levodopa medication level [17,18], behavior [19,20,21] and neurostimulation intensity [8]. Furthermore, LFP signals are stable over long periods of time, as evidenced by recordings from the motor cortex [22] and STN [23,24], a necessary characteristic for a feedback signal in a closed-loop DBS system.

In PD patients, the loss of nigral dopaminergic input to the striatum leads to the symptoms of rigidity and bradykinesia [25] that are related to increased beta frequency oscillatory power (β power). Decreased dopaminergic inputs to the basal ganglia promote synchronized oscillatory activity in the beta frequency (13–30 Hz) of STN-LFP recordings [26,27,28,29] and have been shown to correlate with worsening of rigidity and bradykinesia [16]. Prominent beta frequency oscillations can also be observed from the cortical surface using EEG [17,30] and ECoG electrodes [31]. In concurrence with the reduction of PD motor symptoms, dopaminergic therapy has been shown to suppress synchronized beta frequency oscillations [17,18] and resulted in a new peak in the gamma frequency from 60 to 80 Hz [18] and in the high frequency oscillation of 300 Hz in STN-LFPs [32].

Beta frequency oscillations in the STN-LFP are reduced before and during voluntary movements [21,27,33]. This event-related desynchronization (ERD) in the beta frequency behaves in a manner consistent with movement-related processing in the cortex, specifically the supplementary motor cortex [33]. Beta frequency desynchronization with movement suggests that beta suppression may be a prerequisite of voluntary movement [34,35,36]. Levodopa therapy has been shown to increase the duration and magnitude of relative pre-movement beta frequency ERD [37], strengthening the hypothesis that beta ERD is a non-pathological phenomenon. Event-related synchronization (ERS), or increases in power, occurs in subjects performing voluntary movements in the gamma frequency (>35 Hz) of EEG signals recorded from the motor cortex [38]. This gamma ERS (increased γ power) is felt to represent increased local neuronal computation and thus to reflect normal neural processes necessary for movement. Synchronous gamma frequency oscillations also develop in STN-LFPs and supplementary motor area activity in PD patients only after patients have been treated with levodopa medication (medication “on” state) [17]. As this STN-LFP gamma ERS is only observed in a medication on state, it is assumed to reflect normal neural processes, as in motor cortex.

STN-LFP activity in PD patients is traditionally recorded directly from the macroelectrodes on the DBS lead intraoperatively or postoperatively, in the interval between lead implantation and subsequent connection to the subcutaneous implantable neurostimulator (INS). In these controlled environments, brief recordings are performed with subjects in a reclined operating table or hospital bed and may not be representative of the variable recordings that will be present in a real-world setting. As PD symptoms are dependent on the patient’s attention, alertness and behavior, the STN-LFP activity likely varies with these factors, as well [39,40,41,42,43]. Therefore, inter-day variability will be a challenge that must be addressed in a closed-loop DBS system. In this paper, we chronically record STN-LFP in PD subjects in the medication on and off state while the subject is at rest and performing motor and speech behavioral tasks using a DBS system that is augmented with LFP recording capability [24,44,45]. Reliable chronic recording of neural activity with a fully-implanted system is an imperative first step towards developing a closed-loop DBS system.

## 2. Materials and Methods 

### 2.1. Recruitment

Seven subjects undergoing DBS as standard of care for the treatment of idiopathic PD were enrolled in this study (Table 1). All subjects provided informed consent for participation in this research study, in a manner approved by the HealthOne Institutional Review Board (approval code 418262) and the Food and Drug Administration Investigational Device Exemption regulation (Clinical Trial Number NCT02115802). 

### 2.2. DBS Surgery

Subjects underwent DBS surgery in the off medication state per clinical routine. DBS lead implantation surgery was performed with a Leksell (Elekta, Stockholm, Sweden) stereotactic head frame and Surgiplan (Elekta, Stockholm, Sweden) targeting software. Targeting of the dorsolateral STN was based on a combination of formula-based and indirect coordinates. We used an a priori, formula-based target of (x, y, z): (±12 mm, −2 mm, −4 mm) with respect to the mid-commissural point and anterior commissure (AC)–posterior commissure (PC) plane. Targeting was then adjusted based on indirect targeting from the borders of the red nucleus (RN) for (x, y, z): (3 mm lateral to the RN border, at the anterior border of RN, 2 mm inferior to the superior border of RN). The prescribed sagittal trajectory angle was 60 degrees from the AC-PC plane, and the coronal angle was 15 degrees from a parasagittal plane, with minor adjustments for cortical sulci and blood vessels. Microelectrodes (Alpha Omega, Nazareth, Israel) were positioned in the center and anterior positions or the center, anterior and lateral positions of a BenGun trajectory guide, with a parallel separation of 2 mm. Moderate propofol anesthesia was used without a protected airway only during placement of the burr hole. Clinical microelectrode recording was performed from 15 mm above to approximately 2 mm below the target. Whereas subjects may have had residual effects from propofol during the initial microelectrode recording though thalamic nuclei, we did not proceed with recording in the STN region until patients were fully awake and conversant. After localization of the STN, the microelectrodes were removed, and a permanent 4-electrode DBS lead (Medtronic 3389, Minneapolis, MN, USA) was implanted in the optimal BenGun trajectory. 

In a separate surgery, an implantable neurostimulator (INS) with additional voltage recording capabilities [44] (Activa PC + S, Medtronic, Inc., Minneapolis, MN, USA) was implanted subcutaneously to provide both standard therapeutic stimulation and bilateral local field potential (LFP) recordings.

### 2.3. Data Collection

Simultaneous bilateral recordings were performed in all subjects. All recordings used a subset of the 4 macroelectrodes of each DBS lead. The DBS lead electrode is platinum/iridium, has a surface area of 6.0 mm^2^ and an impedance of 1.7 kΩ (mean; 95% CI = 1.1–2.4 kΩ) [21].

All subjects underwent an intraoperative (OR) data recording session during the initial implantation surgery. Signals were amplified and digitized with a sampling frequency (Fs) of 4.8 kHz (g.USBamp, g.tec, Graz, Austria) and combined with event markers and subject response signals [46]. Skin surface electrodes were used for ground and linked bilateral mastoid common reference. LFP electrodes consisted of linearly-ordered contacts, with 0 being the most ventral and 3 being the most dorsal. Channels were bipolar re-referenced (0–1, 1–2, 2–3) prior to analysis for each brain hemisphere.

Two types of postoperative INS recordings were performed: non-behavioral montage signal review and behavioral dual bipolar recording. During the non-behavioral montage signal review, all bipolar channel pairs from each hemisphere were amplified and digitized with a Fs of 422 Hz by the INS for 30 s sequentially. During the behavioral recording, two bipolar re-referenced LFP signals, one from each hemisphere, were amplified and digitized (Fs 422 Hz) by the INS (Table 2). Using the initial montage signal review, the bipolar pair of channels that contained the most prominent peak in beta frequency oscillations was selected for further behavioral recordings. This selection method emphasized a relative strength of beta power over other frequencies. Simultaneously, the biopotential of the skin over the INS was amplified and digitized (Fs 4.8 kHz) (g.USBamp, g.tec, Graz, Austria) and combined with event markers and subject response signals [46]. The INS was used to produce a 5-Hz, 2-V, 90-µs non-therapeutic stimulation signal at the beginning and end of each recording. Digitized signals were then downloaded from the INS and synchronized to the external signals using the artifacts produced by the non-therapeutic stimulation. Recordings were performed at 1, 3, 6 and 12 months after DBS lead implantation. Montage signal review was performed at each recording session, and behavioral recordings were performed at Months 3 and 6. All postoperative recording sessions were performed in the medication on state, except for the 6-month recording (Table 3). Subject 1 did not have a postoperative recording session in the medication off state. At the time of publication, all subjects have undergone the 1-, 3-, and 6-month sessions, and 5 of the 7 subjects have undergone the 12-month session. Medication off recording sessions were performed after subjects refrained from taking their prescribed levodopa medications for at least 12 h. Medication on recordings sessions were performed while the subjects were administered their prescribed levodopa medication dosage. All recordings were performed with the stimulation off. 

### 2.4. Behavioral Tasks

Behavioral tasks were performed postoperatively and included motor and speech tasks. The motor tasks were cued button pressing, a one out of eight target reaching for both the left and right arm/hand and a cued tongue extension. The speech task was a cued reading aloud of one word presented on a computer screen. Tasks were selected to evoke language- and motor-based neural activity in the STN with precise and consistent timing for robust analysis. More specifically, button press was chosen as a simple motor task that models finger tapping [21,47]. A reach task was selected as an alternative motor task requiring larger muscle recruitment and coordination [48,49]. A brief speech task was chosen because it was consistent in timing with the movement tasks, but similar tasks have been previously used [21,50]. A mouth movement task was used as a comparison to the speech tasks. Each task was repeated with a period of 5 s with a ± 0.5 s jitter in blocks of 11 trials for neurostimulator recordings due to the memory constraints of the INS system with 10 s of reset between each block. The subject was instructed to return to a comfortable resting position between each trial, so the period immediately preceding the cue could be used as a baseline resting state. The entire paradigm was performed up to three times during each session based on subject fatigue. For task initiation, subjects received an audio and visual cue from a presentation computer running a custom Python application (Python Software Foundation, version 3.5, Wilmington, DE, USA). A random time factor (± 0.5 s) was programmed into the trial period length to reduce any effect of anticipation.

### 2.5. Analysis

Time series data of subject response channels were reviewed to mark motor and speech onset and offset times for tasks. Button press responses were extracted from the digital input channel using a threshold algorithm. Reaching motor responses were extracted from the touch-screen monitor digital input channel using a threshold algorithm. Both button and touch-screen inputs were recorded as digital channels; using a threshold of 0.5, values above this threshold (1) indicate a response, while values below this threshold (0) indicate no response. The sensitivities of the button and the touch-screen were left at their factory set defaults. Tongue extension responses were marked from the first time synced video frame the tongue was visible until the last video frame the tongue was visible. Speech responses were manually marked from the start of the first audible syllable above the noise floor until the end of the last audible syllable above the noise floor in the time-synced audio channel.

Power spectral density estimates were calculated using a custom Welch’s method with 50% overlapping Hanning windowed segments of 256 samples padded to 1024 samples for a frequency resolution of 422/1024 (~0.412 Hz). Time-dependent power spectral density estimates were calculated using both wavelet analysis and a fixed-window short time Fourier transform technique [51,52]. Wavelet analysis utilized complex Morlet wavelets and operated on time-by-trial event matrices based on subject response time markers. Button press motor action onset and offset were determined by the digital input channel. For reach motor action, trials were aligned by touch-screen response timing (i.e., at the completion of the reach motion, when the subject hits the target). To measure ERDs, the number of pixels of significant desynchronization in the area of interest around the event were summed. Ratios between two different ERD measures were then computed to compare spectrograms and remove the dependence on pixel density; Student’s *t*-tests were used to compare these ratios against 1 as a significance test.

### 2.6. Statistics

Permutation (bootstrap) analyses were applied to the spectrogram response matrices to determine significant deviation from the baseline of ERS or ERD [21]. These matrices were built using entire-record normalized log transformed data, baselined on a trial-by-trial basis. A time window of 500-ms duration, terminating 100 ms before the cue, was used for baseline data for all tasks. A permutation “sign-test” was performed using randomly-sampled trials without replacement using a custom Python script. For each permutation, the sign of half of the trials was inverted and a new average generated. The null hypothesis states that no ERSs or ERDs are significantly different from zero, and therefore, inversion of the sign would be expected to increase the absolute value of the average for a substantial number of permutations. For the alternate hypothesis, ERSs or ERDs are significantly different from zero, and inversion would be expected to decrease the absolute value of the average.

To correct for multiple comparisons, we conservatively assumed that each 50-ms segment could be considered an independent sample. We calculated the number of permutations, such that the resolution of our *p*-value was 1/5 of the corrected significance value for an alpha error of 5%. This calculation was 5 × (((spectrogram duration)/50 ms)/5%) permutations, which gives corrected *p*-value resolutions of 1/permutations. The spectrogram duration used for statistical analysis and the number of permutations for each task are as follows: button press task, 4 s and 8000 permutations; reach task, 5 s and 10,000 permutations; tongue extension task, 4 s and 8000 permutations; speech task, 4 s and 8000 permutations. To calculate our *p*-value, we counted the number of permutations that created a new mean greater than the original mean and multiplied this count by the resolution of our *p*-value, producing a statistical matrix. If the permutation mean was greater than the original mean, no more than the 5/permutations fraction, the null hypothesis was rejected for that pixel, and it was deemed significant.

The Pearson product-moment correlation coefficient was applied to pairs of calculated power spectral density (PSD) estimations to test for significant spectral correlation between STN-LFP recordings at different times or in different states. Bonferroni correction was applied when testing multiple pairs.

## 3. Results

STN-LFP activity from fourteen hemispheres of seven PD subjects was analyzed. The mean age of the subjects at DBS lead implant was 61 ± 9 years. Two subjects were female. All subjects were evaluated preoperatively using the unified Parkinson’s disease rating scale (UPDRS) III in the medication on and off state. Six of seven subjects were evaluated postoperatively using the UPDRS III in the medication on and off state and DBS stimulation on and off state. Predominant PD symptoms with a notably high value in the initial UPDRS evaluation prior to DBS implantation included bradykinesia, rigidity, dyskinesias, gait disturbances and tremor. Clinical DBS settings from the last recording session can be viewed in Table 1.

STN-LFPs recorded with an external amplifier in the operating room and an INS system in the clinic at six months post lead implantation both yield signals with similar characteristics. Peaks in the beta frequency were visible in 19 out of 36 PSD estimates in the OR recordings and 15 out of 36 PSD estimates in the six-month INS recordings. Beta frequency peaks varied in amplitude and specific frequency range by subject. Figure 1 demonstrates the PSD estimation of representative bipolar channel pairs in the acute and chronic setting for each subject. Although the INS was set at a lower sampling rate of 422 Hz compared to 4.8 kHz (Table 2), the spectral content of the signals was comparable with a characteristic one/frequency curve common in LFP signals. A peak at 60 Hz was visible in two of the six subjects in the intraoperative recording session due to environmental line noise. No 60-Hz artifacts were visible in the postoperative recording sessions with the INS system. Across the full set of bipolar channel pairs, 33 of 36 recordings showed significant correlation between PSD estimations derived from recordings in the acute and chronic setting (multiple correlation tests with Bonferroni correction, α = 0.05/36 (~0.0014)). Channel pairs that had PSD estimations that were significantly different included one channel pair with an impedance consistent with an electrical short (Subject 3) and two channel pairs with a large amount of 60-Hz line noise due to a lack of stable ground connection in the OR recording (Subject 5).

STN-LFPs recorded while each subject was at rest were compared across one year. Similar to the comparison of recording amplifiers, the spectral content in STN-LFP signals varied across subjects, but remained consistent across sessions within subjects (Figure 2). Prominent peaks in beta frequency were visible in 18 out of 36 PSD estimations in the on medication state (Month 3) and 15 of 36 of PSD estimations in the off medication state (Month 6). Peaks in beta frequency varied in amplitude and specific frequency range by subject. The average beta frequency (13–30 Hz) maximum for each subject across all recording sessions ranged from 16.0–25.5 Hz with an average across all subjects of 20.4 ± 4.1 Hz. In early recording sessions, the power analysis subsystem of the INS recording system was programmed to a suboptimal setting that led to a contaminate signal in the recording at the 25, 50 and 75 Hz as seen in PSD estimation for the 1-, 3- and 6-month session for Subject 2, the 1-, 3- and 6-month sessions for Subject 3 and the one-month session for Subjects 1 and 5. This contamination was later avoided by configuring the power analysis subsystem into lower priority frequencies to minimize its impact on the recording. Within subjects, all but one channel pair showed significant correlation between INS PSD estimations over time (multiple correlation tests with Bonferroni correction, α = 0.05/216 (~0.0002))) (Appendix A). The inconsistent channel pair was due to an electrical short in one channel pair in Subject 3. 

STN-LFP activity was further examined for differences between therapeutic and nontherapeutic contacts, as well as differences from the medication on and off state. The spectral content of STN-LFP activity recorded from therapeutic and non-therapeutic contacts while the subjects were at rest in the medication off state at six months was compared (Figure 3). Therapeutic contacts were the active (negative) contact selected by the subject’s neurologist. The therapeutic electrode was deduced after testing each cathode using 60 µs, 130 Hz and selecting the cathode that was most effective in reducing rigidity, bradykinesia and/or tremor with priority given to the efficacious cathode requiring the lowest amplitude to reach the most effective symptom reduction. Side effects were not a factor in selecting the most effective (therapeutic) cathode; however, stimulation-induced side effects were a factor in selecting polarity and pulse width. Clinical DBS settings are listed in Table 1. For Subject 5, a double cathode was used to expand the area of activation when side effects limited the increase in amplitude and pulse width. Therapeutic bipolar channel pairs consisted of at least one therapeutic contact. A prominent peak in beta frequency power was observed in 16 of the 25 therapeutic bipolar channel pairs, although the size and shape of the beta frequency peak was inconsistent across subjects. Notably, eight of 11 non-therapeutic bipolar channel pairs lacked prominent peaks in the beta frequency. Examining only therapeutic bipolar channel pairs across all subjects, there was not a significant decrease in total beta frequency power in the STN-LFP activity recorded at three months post lead implantation with subjects in the medication on state compared to the STN-LFP activity recorded at six months post lead implantation in the medication off state (paired *t*-test with Bonferroni correction, α = 0.05/6 (~0.0083)). Furthermore, there were no significant interactions between therapeutic/non-therapeutic bipolar channels and medication or the laterality of the bipolar channels and medication when measuring total beta power (two-factor ANOVA, α = 0.05). The overall change in total beta power from the on medication to off medication state was −0.28% ± −0.1%. If subjects who exhibited an increase in beta frequency power in the on medication state were excluded, there remained a minimal effect with an average 9% ± 3% decrease in beta frequency power in the on medication state in comparison to the off medication state (Appendix A).

STN-LFP event-related power changes were examined in the medication on and off state at three and six months post DBS lead implantation. All subjects completed six different cued tasks in four categories, left and right button press with a combined reaction time of 0.49 ± 0.18 s, left and right reaching with a combined reaction time of 1.37 ± 0.38, tongue extension with a reaction time of 0.61 ± 0.28 and cued reading with a reaction time of 0.75 ± 0.22. Significant changes in relative power were generally limited to beta (13–30 Hz) and low gamma (35–70 Hz) frequencies, although some significant changes in relative power were visible in frequencies below 13 Hz (Figure 4). The relative ERD of beta and the ERS of gamma frequency oscillations were present in all behaviors, but significant changes in gamma oscillations were not consistently present in all subjects. Gamma ERS was only observed in the medication on state. Dopaminergic medication did not diminish the magnitude of the beta ERD. Across all subjects, dopaminergic medication produced a significant increase in the relative magnitude and duration of beta frequency ERD for all behaviors, except right button press; see Appendix A (paired *t*-test, α = 0.05). Event-related power change patterns over time were variable across behaviors. Left and right arm reach to target movement had the longest duration of relative power change. Unlike the other behavioral tasks, time zero marked when the subject completed the reach at the highlighted target, prior to moving back to resting position. Brief, relative significant changes in power occurred with left and right hand movement. Similar to limb movements, speech and mouth movement showed significant beta ERD and gamma ERS.

## 4. Discussion

In this study, we examine the influence of medication and behavior on STN-LFP features in PD subjects over one year. With the expected ongoing development of closed-loop DBS systems that utilize features of the LFP as a control signal, the stability of this feature modulation in response to patient factors (behavior, medication) is critical to establish. 

The INS system used in this study is an early step in the development of a closed-loop DBS system. The significant advantage of this system is the ability to chronically record STN-LFP. As an implanted system, advantages include the absence of movement artifacts and line noise that contaminate traditional LFP recordings, as seen in Figure 1, and the ability to record LFP signals in a variety of environments and medication states. The INS system utilized in this study is the first of its kind to be implanted in patients [24,44,45]. Two previous studies have utilized the same INS system to examine STN-LFP activity while subjects were performing different behaviors and with varying amplitudes of deep brain stimulation. Although one study was performed in non-human primates [45] and the other in human subjects [24], both studies were able to record reliable LFP signals that were comparable to intraoperative STN-LFP signals. Behavioral recording sessions with the INS are limited in length, number of channels and sampling rate. Recording sessions are stored to an internal memory and retrieved as a second step. The two-step procedure also increases the amount of time that our subjects give to the project. For our recording sessions, we configured the device to record two time-domain channels at Fs 422 Hz without compression, and internal memory restrictions limited recordings to 15 min. Next generation devices may address many of these limitations by increasing the number of simultaneous recording channels and streaming the data in real time, eliminating the recording time limit and decreasing the time needed for each subject visit. 

The INS device used in this study is powered by a non-rechargeable battery that is shared with the neurostimulation circuitry. Thus, utilization of the sensing technology potentially reduces the lifetime of the stimulator. Furthermore, the power analysis subsystem circuitry of the INS system produces a crossover signal in the time domain recording at the configured frequency. By changing the configured power analysis frequency, this contamination can be moved into lower priority frequencies to minimize its impact on the recording. However, our site had noticeable peaks in their PSD at 25, 50 and 75 Hz in early recordings. These artifacts at 25, 50 and 75 Hz did not affect our analysis of beta power at rest or with an event. All our INS recordings were performed with the stimulation off. Active stimulation would create more artifacts in DBS lead recordings, but setting recording parameters, such as the INS recording gain and channel configuration, can limit the influence of stimulation artifact on the recording signal. Furthermore, many groups are examining methods to remove stimulation artifacts [53,54,55] from STN-LFP.

The STN-LFP activity varied between subjects, but was consistent within subjects across recording systems and over a one-year time period. In our study, we found consistent spectral content of LFP activity less than 100 Hz across the one year of recordings. The stability of LFPs has been previously examined up to a span of seven years post DBS implantation. Similar to our findings, beta band amplitude was consistent at three weeks post DBS lead. However, the amplitude decreased at two to seven years after the DBS implant [23]. In the medication on and off states, we observed a prominent peak in beta frequency power spectral density. Beta power from STN-LFP recorded from therapeutic contacts while the subject was at rest did not significantly vary with medication state. Therapeutic contacts were selected for this analysis because they have been found to have higher beta power in comparison to non-therapeutic contacts in previous studies and in our results [56,57]. The suppression of beta frequency activity in the resting state with dopaminergic medication has been observed in other studies [17,18,37], but was inconsistent in our study. 

Similar to previous work, beta ERD was demonstrated to be coincident with a behavioral task [23]. The amplitude of this modulation, relative to a baseline period immediately preceding the cue to start movement, was greater in the medication on state in five of six tasks. This phenomenon, although counter to the pathological hypothesis of beta frequency rhythms in Parkinson’s disease, has been previously observed [24]. This work reinforces the theory that dopaminergic mechanisms facilitate movement-related beta desynchronization. The gamma ERS was, as expected, present only in the medication on state, but was inconsistently observed. However, the frequency of gamma ERS in our study was in a lower gamma range than has been observed in previous studies [32,33,58]. Left and right reaching movements elicited the largest difference between medication on and off states, likely because this behavior involved sustained gross motor movement of the entire arm, where other behaviors were more restricted.

We acknowledge that there are limitations of the current study with respect to our findings on the medication effect on beta power. We demonstrated that there was no significant medication effect on overall beta power present at rest. Furthermore, the on medication state had a significant influence on the strength and duration of beta ERD, when compared to the pre-activity baseline. However, comparing STN-LFP activity during the medication on state at three months to the medication off state at six months would be dependent on a time factor, as well as the medication state. It is possible the recordings at six months could be influenced by a degradation of the signal. We were not able to collect on and off medication data at each three- and six-month encounter. In designing the study, we wanted to preserve the lifetime of the generator and limit the number of recording sessions we performed, while extending the recording to 12 months after DBS implant. We do not believe that our results were due to overall degradation of recording over time. As seen in Appendix A, for the channels chosen for behavioral recordings (Table 1), the prominent beta peak was consistent across 3, 6 and 12 months. Furthermore, the increase in impedance from 3–6 months was minimal at 2.6% ± 18%. We believe this small change in impedance is within the typical range of variability seen in DBS systems [59] and, therefore, demonstrates that the impedances were stable between the two recording sessions. The current study was designed to explore changes in frequency with behavioral tasks, but not to fully elucidate the medication effect. Investigations particularly designed to address this question are planned for future studies.

The magnitude and time course of beta and low gamma power modulation in response to behavior were variable between subjects. This synchronization/desynchronization modulation did not reach significance for certain subjects. Sources for variation between subjects may stem from differences in DBS lead location within the STN, the level of patient’s PD progression, prescribed medication, predominant PD symptoms and unknown factors. This variability has direct implications for the development of closed loop systems. From our work, only a subset of subjects had reliably significant movement-related synchronization patterns for targeting as a control signal for DBS [11]. Strategies for surgical targeting regions of the STN where control signals are optimum may be required for closed loop systems. In this study, surgical targeting was not considered for optimal STN-LFP signals, as the utilization of these signals is not standard clinical practice.

## 5. Conclusions

Our study examined the influence of medication and behavior on STN-LFP activity in PD patients over one year. Many groups are currently investigating appropriate neurophysiological biomarkers to be used as a feedback signal in a closed-loop DBS system [11,49,60,61,62]. The INS system in our study was able to consistently record LFP power fluctuations with behavior, demonstrating an ability to utilize STN-LFP signals as a surrogate for behavioral program activation in a closed-loop DBS system. Although the number of simple behavioral tasks performed by subjects was limited, STN-LFP activity was sensitive to the particular type of motor program. Patterns within STN-LFP signals have been determined to be specific to different motor programs allowing for high classification rates [20]. The data presented demonstrate that STN-LFP activity may be an appropriate feedback signal that is stable over time, providing relevant patient-specific information on a subject’s behavior and medication level.

## Figures and Tables

**Figure 1 brainsci-06-00057-f001:**
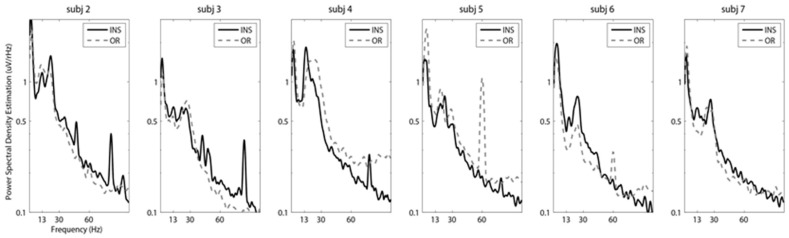
Stability of subthalamic nucleus (STN) local field potential (LFP) signals across recording systems in acute and chronic environments. Representative power spectral density (PSD) estimations of 30-s STN-LFP epochs with subjects at rest in the medication off state are presented. One bipolar channel pair was selected for each subject as an exemplar PSD. The selected bipolar channel pairs for Subjects 2, 3, 5 and 6 were left hemisphere Channels 1 and 2. The selected bipolar channel pairs for Subjects 4 and 7 were right hemisphere Channels 1 and 2. Implantable neurostimulator (INS) digitized signals were recorded at six months post deep brain stimulation (DBS) lead implantation in the medication off state, and STN-LFP signals were recorded in the operating room (OR) during DBS lead implantation. Prominent peaks in the beta frequency were observable in both types of recordings. Subject 1 did not have a recording in the medication off state.

**Figure 2 brainsci-06-00057-f002:**
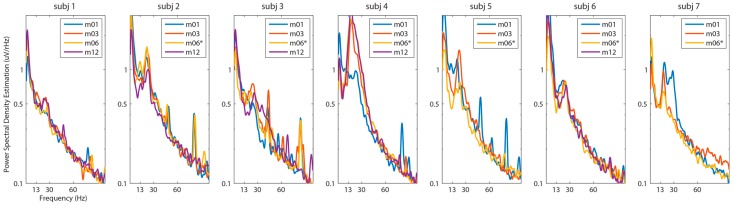
Stability of subthalamic nucleus (STN) local field potential (LFP) signals recorded with an implantable neurostimulator over one year. Representative PSD estimations of 30-s STN-LFP epochs with subjects at rest over one year are presented. One bipolar channel pair was selected for each subject as an exemplar PSD. The selected bipolar channel pairs for Subjects 1, 2, 3, 4, 5 and 6 were left hemisphere Channels 1 and 2. The selected bipolar channel pair for Subject 7 was right hemisphere Channels 1 and 2. Appendix A provides the PSDs for all bipolar channel pairs. STN-LFP frequency content was visibly consistent across multiple recording sessions. Legends denote recording date in months post-surgical implantation * Denotes recording sessions in which the subject was in the medication off state.

**Figure 3 brainsci-06-00057-f003:**
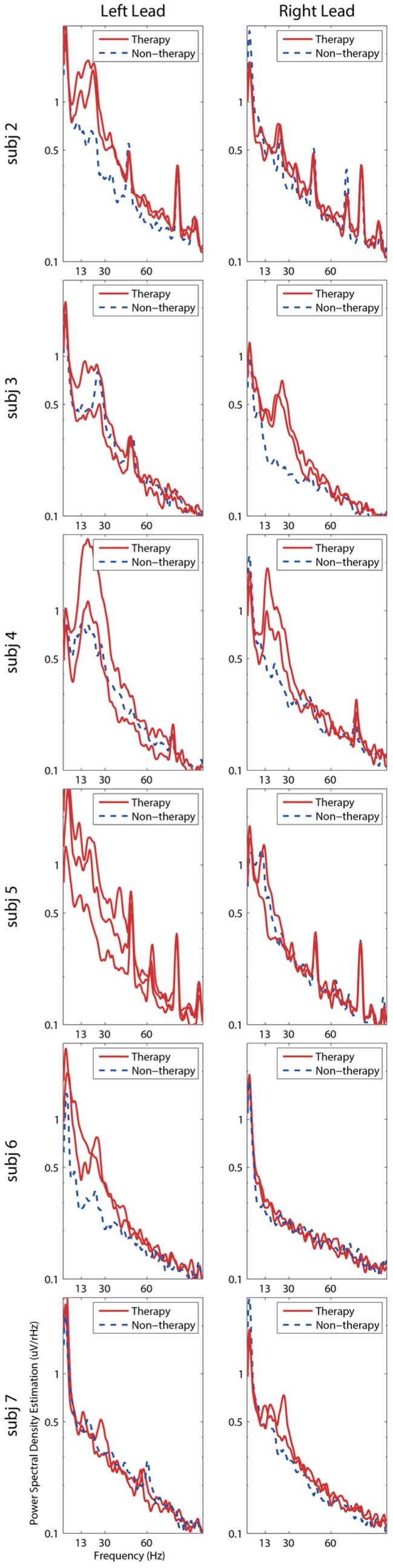
Spectral content of subthalamic nucleus (STN) local field potentials (LFP) signals recorded with an implantable neurostimulator (INS) from therapeutic and non-therapeutic contacts. PSD estimation of 30-s STN-LFP epochs of all bipolar channel pairs for each subject at rest at six months post deep brain stimulation lead implant in the medication off state are presented. Therapeutic bipolar channel pairs consisted of at least one therapeutic (negative) contact for stimulation. Subject 1 did not have a recording in the medication off state.

**Figure 4 brainsci-06-00057-f004:**
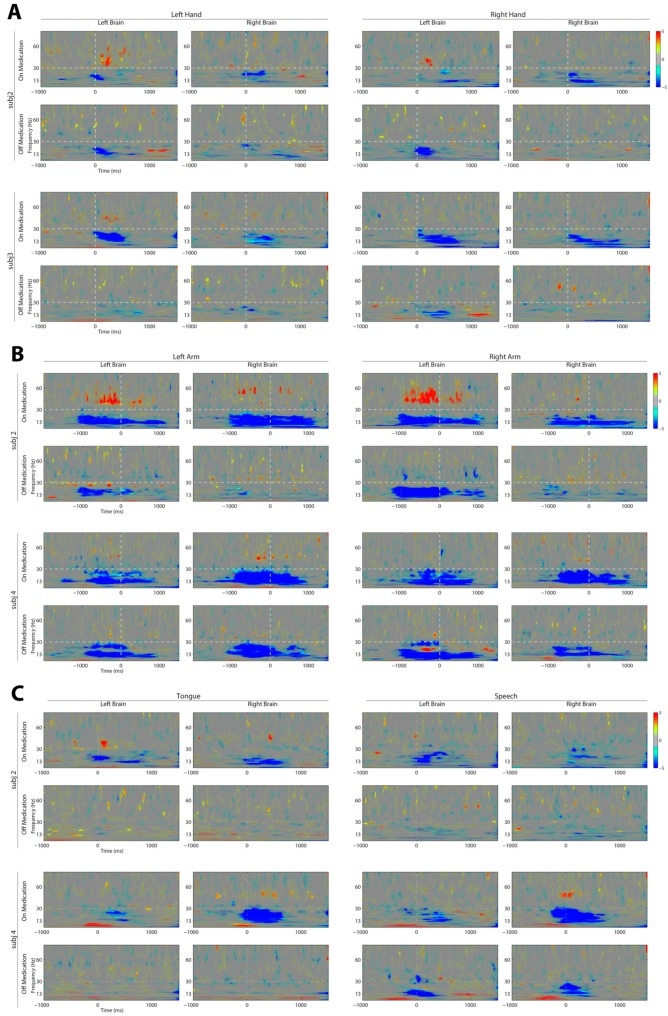
Dopamine-dependent dynamics of subthalamic nucleus (STN) local field potential (LFP) activity recorded with an implantable neurostimulator (INS) with behavioral events. Subjects performed (**A**) left and right button press, Time 0 ms indicates digital response; (**B**) left and right reach, Time 0 ms indicates detection of reaching target by a touch-screen monitor; (**C**) tongue extension, Time 0 ms indicates first observance of tongue with a time-synced video recording; and speech, Time 0 ms indicates the onset of speech as detected by a time synced audio recording. Medication on recordings were performed at three months post DBS lead implant. Medication off recordings were performed at six months post DBS lead implant. STN-LFP event-related desynchronization (ERD) in beta (13–30 Hz) frequencies was observed in the medication off state. STN-LFP ERD in beta frequencies and event-related synchronization in low gamma (35–70 Hz) frequencies were observed in the medication on state. One bipolar channel pair was selected from each hemisphere for behavioral recordings (listed in Table 1). Appendix A provides the spectrograms for all subjects performing each behavior.

**Table 1 brainsci-06-00057-t001:** Subject demographics.

Subject	Age (years)	Sex	Handedness	Pre-op UPDRS III ^1^	Post-op UPDRS III ^2^	Diagnosis and Predominant Symptom	Clinical DBS Settings Left STN	Clinical DBS Settings Right STN	Bipolar Channels for Behavioral Recording
1	59	F	R	37/22	35/24/16/13	PD, bradykinesia	E3⁺, E2^−^, 3.6 V, 60 µs, 130 Hz	C⁺, E1^−^, 3.6 V, 60 us, 130 Hz	L: 2/3, R: 0/1
2	65	M	R	48/14	49/23/17/16	PD, rigidity	E1⁺, E2^−^, 2.6 V, 60 µs, 150 Hz	C⁺, E2^−^, 2.4 V, 60 us, 150 Hz	L: 2/3, R: 2/3
3	63	F	R	23/6	32/25/24/16	PD, dyskinesias	E3⁺, E2^−^, 2.4 V, 60 µs, 130 Hz	E3⁺, E2^−^, 2.5 V, 60 us, 130 Hz	L: 1/2, R: 2/3
4	71	M	R	31/7	38/37/21/20	PD, gait disturbance	E3⁺, E2^−^, 3.6 V, 60 µs, 130 Hz	E3⁺, E2^−^, 3.6 V, 60 µs, 130 Hz	L: 2/3, R: 2/3
5	44	M	R	38/20	36/16/29/11	PD, tremor	C⁺, E1^−^, E2^−^, 2.9 V, 60 µs, 130 Hz	C⁺, E2^−^, 3.2 V, 60 µs, 130 Hz	L: 1/2, R: 2/3
6	62	M	L	31/20	24/16/21/14	PD, tremor	C⁺, E1^−^, 2.6 V, 60 µs, 130 Hz	C⁺, E1^−^, 3.9 V, 70 µs, 135 Hz	L: 1/2, R: 2/3
7	68	M	R	67/40	-/-/-/-	PD, bradykinesia	C⁺, E1^−^, 2.2 V, 60 µs, 130 Hz	C⁺, E2^−^, 2.0 V, 60 µs, 130 Hz	L: 2/3, R: 2/3

^1^ Pre-op UPDRS III listed in order of Medication off/Medication on; ^2^ Post-op UPDRS III listed in order of Medication off-DBS off/Medication off-DBS on/Medication on-DBS off/Medication on-DBS on. PD, Parkinson’s disease. UPDRS, unified Parkinson’s disease rating scale. STN, subthalamic nucleus. M, male. F, female. L, left. R, right.

**Table 2 brainsci-06-00057-t002:** Comparison of amplifier settings for intraoperative and postoperative recording sessions.

	OR	INS
**Input impedance**	>100 MOhm	1 MOhm
**Range**	−250 mV–250 mV	−10 V–10 V
**Filters used**	0.5–2000 Hz	0.5–100 Hz
**Sampling Rate**	4800 Hz	422 Hz
**Noise Floor**	<0.3 µV RMS (0.1–10 Hz)	Min signal to detect 1 μV RMS differential with noise floor <0.3 µV RMS

OR, operating room. INS, implantable neurostimulator. MOhm, megaohm. RMS, root mean square.

**Table 3 brainsci-06-00057-t003:** Timing and type of recording sessions performed.

Recording Session	Intra-Operative	1 Month	3 Month	6 Month	12 Month
**Recording Type**	OR	INS	INS	INS	INS
**Medication State**	Off	On	On	Off	On
**Non-behavioral Montage Recording**	6 bipolar channels	6 bipolar channels	6 bipolar channels	6 bipolar channels	6 bipolar channels
**Behavioral Recording**	-	-	2 bipolar channels	2 bipolar channels	-

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
