# Peer review of "Long-Term Task- and Dopamine-Dependent Dynamics of Subthalamic Local Field Potentials in Parkinson’s Disease"

_brainsci, 2016, doi:10.3390/brainsci6040057_

Round 1

Reviewer 1 Report

Summary

This study analyses effect of dopaminergic medication on STN LFP beta rhythms during rest and motor task periods. This effect is compared in short and long term recordings. For intraoperative recordings gTec amplifier with a fs:4.8KHz and for the long term recordings an implantable neurostimulator (INS) with fs:422Hz is used. Overall 7 subjects with bilateral DBS electrode implantation (14 STN nuclei in total) are included in the study. The novelty here is that this study is one of few to use the INS in human subjects (might be the first for PD)

Study presented can be divided into 4 sub-sections:

1. Validation of LFP from INS with intraoperative recordings from gTec

2. Robustness of LFP in long term up to 1 year (1,3,6,12 months) 

3. Beta changes during rest with medication (all bipolar contacts)

4. ERD and ERS comparison during behavioral tasks when med is OFF and ON (bipolar contacts with strongest beta)

Authors claim that INS can record similar beta to intraoperative recordings obtained with gTec (1), and LFP characteristics are sustainable over long term. They also show that during rest period, the amplitude of beta did not diminish over a year. They say there is 9% decrease over 6 months but it is calculated by excluding 3 of 6 subjects. The other 3 has increase in beta. Also, they report that reduction of beta power from therapeutic electrodes with dopaminergic medication was inconsistent in their study. Parallel to previous studies, they note that beta ERD is stronger when med is ON and gamma ERS is only present when med is ON. A summary of major outcomes are as follows.

1- Intraoperative LFP and INS has similar characteristics within patient. Thus, INS is sufficient to be used for LFP analysis.

2- LFP is shown to be a robust signal with consistent beta peaks over time.

3- Beta energy did NOT diminish with dopaminergic medication (as opposed to previous studies).

4- Beta ERD is stronger with med ON, gamma ERS is only present when med is ON (as expected from older studies)

They conclude that STN-LFP has distinguishing features for different motor tasks as well as medication states. Besides, it is long-term reliable. Thus, it can be appropriate feedback signal for a closed-loop DBS system.

This study is important since it represents an early clinical application of INS to study STN-LFPs. However it suffers from a number of issues such as poor study design and the conclusions listed above are not supported by their data very well. 

Major concerns related to the study are listed below:

1- One of the major aims of this study is to show the robustness of LFPs over long term recordings. This has been shown previously by Abosch et al (2012) in the article “Long-term Recordings of Local Field Potentials” which is surprisingly not cited in this manuscript. The group compared LFP signals up to 7 years span which is much longer than 1 year span of this study. The results obtained in the current should be compared to Abosch et al. and disussed.

2- The critical details about signal processing techniques (163), such as frequency resolution of PSD estimates or window sizes and types, should be provided in order for reader to have a better understanding of the results. The spectra in figures have unrelated small ripples which make them look over-crowded. If window size, overlap and FFT resolution parameters are modified, a cleaner/smoother figure can be obtained without compromising the prominent peaks in frequency bands of interest.

3- It is not clear why there are 30 electrode pairs (216). It should be 3x6x2 according to info provided in “data collection” section of the article. 

4- Statistical methods for some analyses are vague. E.g. how does authors compute the correlation of PSD estimations (217)? Which characteristics did they compare: morphology or band power or both? Also level of significance is not provided for correlation between PSD estimates at (217) whereas an alpha value is given for the other correlation analysis (240). The presentation of results should be consistent.  

5- PSDs given for 6th month recordings in Figure 1 (222) and Figure 2 (242) are not consistent. This is a major problem since they differ in location of beta peak as well as its amplitude.

6- The problem with INS system and the way it was fixed is not clear (238). From the explanation, it is inferred that the contamination of 25, 50 and 75 Hz is still present but less in amplitude due to selective filtering of lower frequencies. However, 25 Hz lies right in the middle of beta range. How come this artifact did not affect your analysis (349)? An explanation of removal of this artifact can be included in paper to show validity of the analysis.

7-“Therapeutic and non-therapeutic electrodes” (249) are confusing. Are these therapeutic electrodes or bipolar contact pairs of the same electrode? Electrode terms user interchangeably along the manuscript . The article does not imply different placement locations. Also the definition of therapeutic is given as “maximum benefit, minimum side effect”. This is not always the case. Which one is prioritized? If one pair has maximal benefit and another has minimal side effect, which pair would the neurologist pick as therapeutic?

8- The authors should provide a ratio/number instead of phrases such as “majority/most” when describing the occurrence of beta peaks in PSD estimations (254). 

9- What is the motivation behind the statistical analysis for comparison of 3-month med ON with 6 month med OFF? (257-263). In one case med is ON and LFP is used intraoperaively immediately after implantation. In the other case med state is OFF and the LFP is recorded after long period following implantation. The decrease of beta cannot be attributed to a single factor here. The 3-month recording might have had higher amplitudes of beta but suppressed due to med ON, whereas by the 6th month the beta amplitude diminished to a similar level by itself due to degradation.

10- The results on beta power reduction with med OFF-ON (262-263) may not be reliable as 3 of the 6 subjects have increase. How can the authors say that there is 9% decrease overall with excluding half of the data set which have increase?

11- In Figure 3, (279) the legend has 2 entries whereas each figure shows 3 plots. Assuming that therapeutic electrode refers to therapeutic contact pair, then does every patient have at least 2 sets of therapeutic contact pairs? E.g. subj.5 has 3 solid lines and no dashed lines. This is confusing, please clarify. Also color plots similar to Fig.2 might be more suitable to represent this data.

12- “Discussion” section (297-366) has a different order than the rest of the article. “Results” has started with comparison of intraoperative recording system with INS. Then it moved to analysis of LFP over time and was finalized with behavioral comparison. However in “discussion” section, behavioral tasks are commented on first, then concerns about INS were explained and the section concluded with comments on long term LFP. To have integrity of flow, paragraphs in “discussion” can be reorganized. 

Minor Issues:

1- The sentence “Beta frequency desynchronization with movement suggests that beta suppression may be a prerequisite of voluntary movement.” (61-62) sounds like a conclusion rather than intro since no citation provided. Based on what, do the authors suggest this?

2- There is a claim (77-78) that PD symptoms are dependent on some factors such as attention, alertness and so on. Yet no citation presented. This claim leads authors to make the conclusion of inferiority of OR recordings which is later (207-208) found not to be true as INS and intraoperative recordings yielded fairly consistent signals. 

3-Although the article has a “subjects” section (85), no information about subjects provided here. Instead authors presented relevant information in the “results” and Table.1. The subject info should be moved to the appropriate section intended for it since readers will be able to know where to find it. If it is necessary to provide the same information in the “results”, at least an in-text reference should be provided in between sections.

4-Adding “Sampling Frequency (or FS):” 4.8 KHz might increase clarity for not-very-technical readers (118,124).

5-Although there is a convention, it is advisable to specify what 0,1,2,3 (120) means in terms of contact positions on the electrode. E.g. dorsal-ventral or top-bottom as this might be relevant for others who would like to benefit from this work.

6-The information about the recording (medication state, behavioral & non-behavioral tasks and recording times) can be added as a column/columns to table 1 for each patient (122, 133-138). It is particularly difficult to keep this multi-faceted information in mind. Using a tabular presentation, the reader will not have to read “methods” section over and over.

Author Response

See word document

Reviewer 2 Report

Summary:

This study examines the reliability of INS chronic recordings from STN of PD patients. They compared PSDs corresponding to INS and OR recordings during rest and across different movement and speech tasks. Evaluating INS recordings based on expected behavioral and pathological related features is a crucial step towards designing real-time closed-loop DBS systems.

There are major concerns in the design and analyses. Mainly:

1-      Other than doing different movements and behaviors, the rationale behind task selection is not clear and seems arbitrary.

2-      Using maximal beta peak power to analyze (rather than using the therapeutic contact) lacks appropriate justification. It is not clear why authors used beta peak and not total beta power for example? Or why they didn’t select the therapeutic contact pair for their longitudinal analyses.

3-      Although the authors claim to study interaction between movement/behavior and medication, this study lacks appropriate statistical analyses of interactions. Moreover, the authors fail to report how the interaction between the two changes across time.

Comments specific to each section:

Introduction:

·         Line 17: very vague sentence: how are STN LFPs accessible? what behavioral or pathological states do you mean by this description? Is this related to PD?

·         Line 23: “voluntary behaviors” is a non-specific phrase. What kind of behavior do you mean?

·         Line 25: gamma (35-70 Hz) is technically low gamma. Please correct throughout.

·         Line 39: in the part “,but DBS may lead to ..” word “DBS” is redundant.

·         Line 40-43: for sentence “An adaptive DBS … symptoms” you need to provide references justifying the claim

·         Line 43-44: sentence “Local field potentials (LFP) recorded from the accessible STN have the potential to be arobust control signal to indicate change in a patient’s state” needs a reference

·         Line 46-48: “ Furthermore, LFP signals are stable over long periods of time, as evidenced by recordings from the motor cortex [14], a necessary characteristic for a feedback signal in a closed-loop DBS system.” You should also reference Quinn et al 2015 using Medtronic Active PC+S extended recordings showing consistency of STN-LFP recordings.

·         Line 49-51: beta power changes with respect to tremor are different. Resting tremor is associated with beta power decrease in the STN signals. The same concern applies for line 53 “many PD motor symptoms” this should be more specific (only rigidity and bradykinesia and not tremor)

·         Line 57: 60-80 Hz (low gamma) is different from 300 Hz (High frequency oscillations) please separate the two. Selection of 60-80 Hz (low gamma) band seems to be motivated/limited by the noise floor of the ACTIVA PC+S (roughly <80 Hz). If that’s the case please include that in your justification.

Materials and methods:

·         Data Collection:

o   Line 126: Seems like the authors used largest beta peak as a selection criteria for bipolar recordings. One might wonder how different this selection was from the contacts for clinical stimulation. Also authors didn’t provide any justification on how they chose beta peak and not the total beta power for their channel selection. Was there any difference observed? With the emerging literature on the existence of two beta sub-bands (low and high [van Wijk, et al 2016]) they should also provide information on how different the peak frequency was across the 7 subjects. A table describing all the details for all subjects will be informative. Without proper justification, channel selection seems arbitrary. Also one might wonder if the channel with largest peak changes over the time for 1,3,6 and 12 months recordings, if the montage review was done, details should be reported. so this needs to be reported as one of the limitations of current study related to the device (Activa PC+S). Also how long was the montage recording and how did the peak detection was performed? Very unclear

o   Line 140-142: medication dosage should be described in a table

·         Behavioral tasks:

o   What is the rationale for using each of the tasks described? What are the specific questions each one of the tasks/experiments were trying to address?

o   What are the timing parameters associated with each task? Details are not clear

·         Analysis:

o   Line 158-160: How much variability in timing was observed across the cohort and trials?

o   Line 155-158: how does the threshold algorithm work? How sensitive it is? Please provide a brief description of the method

o   Line 163: power spectral density: please provide frequency resolution and type of windowing (Hanning?).

o   Line 173: “A time window of 500 ms duration, terminating 100 ms before the cue was used for a baseline data for all tasks”. For the reaching task, doesn’t this baseline period contain some amount of movement activity? Literature supports changes occurring before overt movement.

o   How long is the duration of spectrograms for each behavioral task (for statistical analysis)?

·         Results:

o   Table1: subject 3 and 4, how long before the surgery the pre-op UPDRS was evaluated for these subjects? Their pre-op on Med UPDRS is lower than their post op UPDRS (even with stim on, meds on)? Also speaking of predominant symptom, was this symptom the symptom subjects initially came for diagnosis or is this reflective of their UPDRS score? Please clarify

o   Was UPDRS assessed during the course of the year? Was there any change? You should report and justify that disease state remained the same using UPDRS. That might be also helpful in justifying why for some subjects you see difference in PSDs over time (Figure 1).

o   Line 209: how many PSD showed visible peaks? How much frequency variability was observed? Was there always a single peak or multiple peaks? From Figure 1, it seems there are two distinct peaks

o   Figure 1: you say you analyzed lfp from 14 STN/7 Subjects but you show only 7 PSD in this figure. How did you choose? Did you average between left and right side? Please clarify.

o   Figure 1: from figure 2, it seems like subject 1 didn’t show any spectral peak in the beta range. How was the channel pair selection done for this subject then?

o   Figure 2: subjects 4, 5 and 7, m01 recording seems very different from other recordings, peak frequency also seems to be variable across the recordings for these subjects

o   Line 239: where are the results of correlation analysis? A table with correlation coefficients and corresponding p-values should be included given the visual inspection of figure 2 provided earlier

o   Line 254-256: please specify how many of the STN/subjects examined showed peaks. “majority” seems very descriptive and not specific enough.

o   Line 249: How did you choose between two active contacts for left STN in subject 6?

o   Line 258: when you say “beta frequency power” do you mean total beta power? Or just the power of the beta peak? Why do you compare beta power at 3 mo ON medication vs 6 mo OFF medication? Please quantify the difference once rationalized (what was the p-value?)

o   Line 262: did you compare on and off medication data at the same time point (ie 3 mo or 6 mo?)

o   Figure 3: at what time point were these recording done? The legend doesn’t show which one of the solid black lines represent on or off medications?

o   Line 249-263: If I understand correctly the point of this specific analysis was to address the difference in beta power between therapeutic and non-therapeutic contacts ON/OFF medications. Was any interaction analysis (2 way ANOVA or mixed effect modeling) done to assess if ON/OFF medication beta power differences was significantly different for the therapeutic contact vs non-therapeutic contact? Seems like some subjects show beta power increase (excluding subject 1, subject (2,4,7), did this happen for both left and right STN?) while other showed increase? This means half your data showed increase and the other half showed decrease in beta power – which may be a random result.  Was this pattern consistent at the single subject level (meaning over time subject showing medication related beta power increase always show similar change at 1,3,6 and 12 months time points?)

o   Line 264-278: at what time point was this analysis done? Was the relationship described consistent across time (ie 1,3,6 and 12 months?) not clear from the text or Figure 4.

o   Figure 4. Each panel show one task (A: button press, B: reach, C: tongue extension) only for a representative subject. One needs to also assess the variability across the subjects/show an average behavior across the cohort.  How was the axial task (tongue extension) was different from lateralized tasks (button press and reach)? Were the ERD and ERS patterns described any different? (for example did the STN side ipsilateral to the movement side, show difference in ERD and ERS compared to the contralateral STN? And how was the relationship for speech and tongue task?)

·         Discussion

o   Line 298-299: authors claim to report influence of medication and behavior on STN-LFP in PD over the one year time course. However based on the results reported, it is not clear whether:

o   There was any difference between different tasks and their interaction with medication (lack of appropriate statistical analysis, investigating interaction between fixed effects)

o   It is not clear at what time point (surgery, 1,3, 6 or 12 month) the analyses were done. And if there is any time dependent changes in the interactional effects

o   Please include an overview of current studies with ACTIVA PC+S. for example: Quinn et al 2015 (Movement disorders) reported a similar study across different behaviors (sitting, standing, etc) with similar recordings across time. Please include that study in your discussion and compare your analysis with theirs.

o   Please include a section on limitation of the study, such as device related limitations.

Author Response

See word document

Round 2

Reviewer 2 Report

While the data is imperfect, the authors have adequately addressed concerns. The material contained in the manuscript is valuable to the community.